# Associations of IL-18 with Altered Cardiovascular Risk Profile in Psoriatic Arthritis and Ankylosing Spondylitis

**DOI:** 10.3390/jcm11030766

**Published:** 2022-01-30

**Authors:** Krzysztof Bonek, Ewa Kuca-Warnawin, Anna Kornatka, Agnieszka Zielińska, Małgorzata Wisłowska, Ewa Kontny, Piotr Głuszko

**Affiliations:** 1Department of Rheumatology, National Institute of Geriatrics, Rheumatology and Rehabilitation, 02-637 Warsaw, Poland; agnieszka.zielinska@spartanska.pl (A.Z.); mwislowska@wp.pl (M.W.); zruj@mp.pl (P.G.); 2Department of Pathophysiology and Immunology, National Institute of Geriatrics, Rheumatology and Rehabilitation, 02-637 Warsaw, Poland; ewa.kuca-warnawin@spartanska.pl (E.K.-W.); anna.kornatka@spartanska.pl (A.K.); ewa.kontny@wp.pl (E.K.)

**Keywords:** cardiovascular risk, psoriatic arthritis, dyslipidemia, IL-18, spondyloarthropathies

## Abstract

Objective: To investigate the associations of IL-18 serum levels with serum lipids, cardiovascular risk, and disease activity in patients with ankylosing spondylitis (AS) and psoriatic arthritis (PsA) with axial (axPsA) and peripheral (perPsA) joint involvement. Methods: 155 adult patients (PsA 61/AS 94) were enrolled in the study. Standard disease activity indices, BASDAI, and ASDAS, were calculated for AS and PsA and DAPSA for PsA. Sera from peripheral blood samples were obtained after night fasting. Serum concentrations of cytokines (IL-18, IL-17) were measured by ELISA, while lipid profile with total cholesterol (TC), triglycerides (TG), low-density cholesterol-(LDL), high-density cholesterol (HDL), and C-reactive protein (CRP) concentrations were determined using routine procedures. The atherogenic index was calculated using the standard formula AI = TC/HDL. Results: Patients with PsA and peripheral joint involvement (perPsA) had significantly higher IL-18 serum levels than axial PsA and AS patients (medians 160 vs. 116 vs. 80 pg/mL). In patients with PsA and in the subgroup with PsA+ ischemic heart disease (IHD), IL-18 positively correlated with atherogenic index (AI) (rho = 0.46 and rho = 0.67, respectively) and TG serum concentrations (rho = 0.4 and rho = 0.675), while negatively with HDL levels (rho = −0.37 and rho = −0.608). In PsA + IHD subgroup IL-18 serum levels correlated positively also with disease activity (DAPSA) (rho = 0.613). Importantly, in patients with perPsA, characterized by the highest IL-18 serum levels, cardiovascular risk, and frequency of both hypertriglyceridemia and IHD, positive correlations between IL-18 and IL-17 (rho = 0.47, *p* = 0.002), TG (rho = 0.45 *p* = 0.01) levels and AI (rho = 0.63 *p* = 0.021) were found. Whereas linear regression models revealed that IL-17, TG concentrations and the tender joint count had an impact on IL-18 Conclusions: We confirmed that patients with perPsA are characterized by a more pronounced proinflammatory and proatherogenic cardiovascular risk profile than patients with axPsA and AS. Importantly our study indicates that in PsA, but not in AS, elevated serum concentration of IL-18 is associated with higher disease activity and proatherogenic lipid profile, leading to a higher cardiovascular risk. Thus, our results point out IL-18 as a critical contributor in these pathological processes and possible therapeutic targets.

## 1. Introduction

Psoriatic arthritis (PsA) is a chronic condition associated with psoriasis (PsO), spondylitis and peripheral arthritis, and comorbidities, including cardiovascular disorders with an increased risk of death [1,2,3]. PsA has been classified as a form of spondyloarthritis (SpA), and it shares some similar proinflammatory pathways with other SpAs [4,5,6,7] and similar metabolic burden with psoriasis [7,8]. However, PsA seems to have a different cardiovascular risk profile and higher cardiovascular risk (CVR) than other SpAs and cutaneous psoriasis (PsO) [1,2]. So, one can assume that cardiovascular risk in PsA is not only a sum of risk factors associated with PsO and SpA but may result from different pathogenic factors. To enhance the assessment of cardiovascular risk in this group of patients, it should be considered that disease activity, psoriasis, arthritis, metabolic syndrome development, and atherosclerosis are intertwined, although data on the nature of their relationship is uncertain [7,8,9]. In search for PsA dedicated proinflammatory mediators matching the aforementioned conditions, selected cytokines, including IL-18 and IL-17, known to be implicated in SpA pathogenesis, were investigated. IL-18 seems to have a special role since IL-18 gene expression was found in psoriatic plaques, osteoblasts, synovial fluid, and atherosclerotic plaques [8,9,10,11,12,13,14]. So far, the relationship between atherosclerosis, blood lipids, glucose, joint inflammation, and IL-18 seems to be complex and not entirely clear [8,9,10,11,12,13,14]. According to some recent reports, high serum levels of IL-18 in PsA are associated with arthritis and skin inflammation [12,13] and with an increased level of IL-17 [14], an interleukin playing a key role in the pathogenesis of axial of SpA [15,16]. There are also reports indicating that increased expression of IL-18 [17] and IL-17 in arterial endothelium may lead to the rapid development of atherosclerosis [14]. In addition, IL-18 may link systemic inflammation with pathological bone remodelling in arthritis due to induction of osteoclast formation and accelerated bone resorption [17,18,19,20,21,22]. Therefore, it is justified to undertake research to verify the hypothesis about the role of IL-18 as a potential “keystone” linking increased cardiovascular risk with active inflammation in PsA. Furthermore, we have investigated whether IL-18 could be responsible for the differences in cardiovascular risk profiles between PsA and AS.

## 2. Materials and Methods

### 2.1. Patients

Two groups of patients who fulfilled the criteria for ankylosing spondylitis (*n* = 94) and psoriatic arthritis (*n* = 61) were recruited for the study. The diagnosis was based on the criteria for the Classification of Psoriatic Arthritis (CASPAR) [23], and the diagnosis of AS was based on the Assessment of Spondyloarthritis International Society (ASAS) criteria for axial spondyloarthritis with X-ray changes [24]. Exclusion criteria included overlapping symptoms or diagnosis of other rheumatic diseases, infections, history of malignancies, acute myocardial infarction, and stroke. Patients treated with biologics like TNF and IL-17 inhibitors, or receiving glucocorticosteroids (GS) exceeding 15 mg of prednisone (or equivalent) per day, were not included in this study. The concentrations of IL-18 and IL-17 were measured in the sera of patients and control group, which comprised 38 age- and sex-matched healthy volunteers.

### 2.2. Methods

Sera were collected from peripheral blood samples after an overnight fast. The lipid profile, i.e., total cholesterol (TC), triglycerides (TG), low-density cholesterol (LDL), high-density cholesterol (HDL), as well as C-reactive protein (CRP) concentrations in sera samples, were determined using routine procedures. The atherogenic index was calculated using the standard formula AI = TC/HDL. Non-HDL cholesterol was calculated using the standard formula Non-HDL-C = TC-HDL. Data on cardiovascular risk factors and ischemic heart disease (IHD) were obtained from the patients’ medical history. None of the patients underwent cardiosurgical intervention. Patients with the confirmed occurrence of atherosclerotic plaques during the percutaneous coronary intervention (PCI) were classified as IHD positive group. Patients with medical history suggesting IHD without PCI confirmation were excluded from the study. Patients diagnosed with IHD preceding the diagnosis of PsA or AS were not included in the study. All patients underwent physical examination, and their weight, height, waist, and hip ratio were measured. Bath Ankylosing Spondylitis Activity Index (BASDAI), Bath Ankylosing Spondylitis Functional Index (BASFI), and Health Assessment Questionnaire (HAQ) questionnaires were filled out by all patients. Disease Activity in Psoriatic Arthritis (DAPSA), Ankylosing Spondylitis Disease Activity Score (ASDAS), and Bath Ankylosing Spondylitis Metrology Index (BASMI) were calculated according to standard formulas. The study was approved by the Ethics Committee of the National Institute of Geriatrics, Rheumatology, and Rehabilitation, Warsaw, Poland (approval N° KBT-8/4/2016). All patients have fulfilled written consent to participate in the study.

### 2.3. ELISAs

Sera were isolated by routine laboratory methods, and samples were stored in aliquots at −70 °C until assayed. Cytokine concentrations were measured using Human IL-18 ELISA (Invitrogen, BMS267-2) and Human IL-17 ELISA (Abcam, ab100556) kits. The assays were performed according to the manufacturer’s protocols. All measurements were carried out in duplicates.

### 2.4. Statistical Analysis

Statistical analysis was performed using the PS IMAGO v 27 SPSS module. The Shapiro–Wilk test for the normality of data distribution was performed for all measured parameters. Data are shown as the median with interquartile range (IQR) for normal and non-normal variable distribution. Parametric (Pearson’s linear) and nonparametric (Spearman’s rank) correlation tests were used to assess an association between tested parameters. For comparing numerical data, we have performed t-Student’s tests for normally distributed data and Mann–Whitney test for non-normally distributed data. For qualitative data analysis, Chi^2^ and Fisher’s exact test were used. The linear models were built using numerical data with Aikake informative criterion (AICc) for all models, coefficient of determination (adjusted R2) to measure strength of the relationship between the model and dependent variable, and Aikake weights (wi) for model selection with goodness-of-fit of presented models [25]. Regression equations are presented according to American Medical Association (AMA) standard. Probability values less than 0.05 were considered significant.

## 3. Results

Demographic and clinical characteristics of AS and PsA patients are presented in Table 1.

### 3.1. Higher Proatherogenic Profile and Serum IL-18 Levels in PsA Than AS Patients

As shown in Table 1, there were no significant differences between PsA and AS patient groups in age, disease duration, erythrocyte sedimentation rate (ESR), and CRP values. Interestingly, CRP and ESR medians were mildly above the normal ranges. Our observation stands in line with current research. It has been reported that despite low or normal values of CRP in AS and PsA, the disease can be considered active. Recent studies, such as Sokolova et al. [26], presented that, similarly to our study, the inflammatory mediators’ expression pattern is mostly related to the disease phenotype. Additionally, in the study by Brown et al., 110 out of 197 patients with moderate to severe AS enrolled into MEASURE 1 and MEASURE 2 studies had lower CRP than 10mg/L [27].

Both groups of patients also had similar disease activity measured as BASDAI; however, AS patients had higher ASDAS-CRP scores than PsA patients with the axial disease (3.6 vs. 3.1, *p* = 0.011). Patients with AS were more often treated with nonsteroid anti-inflammatory drugs (NSAIDs) than patients with PsA. Conventional synthetic disease-modifying antirheumatic drugs (DMARDs), especially methotrexate, were used more frequently in PsA patients. Compared with AS patients, the PsA group was characterized by significantly higher serum triglyceride (TG) concentration, higher fasting glucose levels, and a tendency towards obesity (higher BMI, waist and hip circumferences) but slightly lower HDL concentration (Table 1).

When compared with healthy volunteers (*n* = 40), PsA (*n* = 18) patients revealed higher IL-18 levels (means: 324 pg/mL vs. 164 pg/mL) and similar IL-17 (26 vs. 20 pg/mL *p* > 0.05) serum concentrations. In AS (*n* = 22) patients group IL-18 concentrations were comparable to healthy controls.

Comparison of patients groups (AS, *n* = 94 PsA, *n* = 61) in terms of serum cytokine concentrations revealed comparable IL-17 but significantly higher IL-18 levels in PsA than AS groups (medians 140 vs. 80 pg/mL, *p* = 0.026) (Table 1). Importantly, in PsA group IL-18 concentration positively correlated with AI (rho = 0.46, *p* < 0.001), and TG concentration (rho = 0.4, *p* < 0.001) and showed weak negative correlation with HDL levels (rho= −0.37, *p* = 0.004) (Figure 1A–C).

We did not observe similar relationships in the group of AS patients (Figure 1D–F). Similar yet stronger correlations were observed while analyzing a small group (*n* = 14) of patients with PsA and concomitant ischemic heart disease (Figure 2A–C).

In this group of patients IL-18 serum concentrations strongly and positively correlated with TG levels (rho = 0.675, *p* = 0.001), AI (rho = 0.671, *p* = 0.004), and a negative correlation between IL-18 and HDL (rho= −0.608, *p* = 0.03) was noted. In this subgroup IL-18 levels correlated positively also with disease activity-DAPSA (r = 0.613, *p* = 0.023) (Figure 2D).

### 3.2. perPsA Patients Are Characterized by the Highest Cardiovascular Risk, Frequency of Hypertriglyceridemia and IHD, and IL-18 Serum Levels

PsA group of patients was divided according to clinical phenotype into two subgroups, patients with the axial disease (axPsA, *n* = 14) and patients with peripheral joint inflammation (perPsA, *n* = 47). Cardiovascular risk factors for these subgroups are shown in Table 1 and Table 2. Our data indicate that perPsA patients suffered more frequently from ischemic heart disease than other groups of patients (AS 9.6% vs. axPsA 0.00%vs perPsA 29.8%) (Table 2). Patients with perPsA presented a tendency towards obesity (AS 17% vs. axPsA 21.4% vs. perPsA 42.6%), revealed dyslipidemia (Table 1 and Table 2), and more often hypertriglyceridemia (AS 16.00% vs. axPsA 14.3% vs. perPsA 48.9%), compared with other groups of patients. It is worth noting that patients with perPsA had significantly higher IL-18 concentrations than patients with AS and axPsA (medians 80 vs. 118 vs. 160 pg/mL) (Table 1).

### 3.3. Associations between Clinical Data, CVD Risk Factors, and Serum Cytokine Levels in perPsA Patients

In the perPsA group (*n* = 47) IL-18 levels correlated positively with TG (r = 0.45, *p* = 0.01), non-HDL cholesterol (r = 0.35, *p* = 0.006), the number of tender joints (TJ) (rho = 0.48, *p* = 0.017) and AI (r = 0.63, *p* = 0.021) but negatively with HDL levels (r = −0.43 *p* = 0.023) (Figure 3).

We also observed positive correlations between IL-17 and IL-18 (rho = 0.47, *p* = 0.002), In linear regression models for the perPsA group, including lipid profile (LDL, HDL, TG, TC), clinical data (occurrence of enthesitis, dactylitis, uveitis are presented in Appendix A), treatment (with methotrexate, sulphasalazine, cyclosporine, NSAID, and glucocorticoids), and disease activity indices (CRP, ESR, DAPSA, ASDAS-CRP) we have found that IL-17, TG concentrations and tender joint count are associated with IL-18 level (Table 3). Comparison of three linear regression models, based on the highest wi and AICc, revealed that TJ, TG, and IL-17 have a major impact on IL-18 levels. Interestingly, despite small delta AICc between models, the highest wi was found in the model including IL-17. Such discrepancy suggests a major impact of IL-17 of IL-18 pathway, yet its direct influence on IL-18 concentrations seems to be minor. Multiple linear regression was calculated to predict IL-18 concentration based on IL-17, TG levels, and TJ. Preliminary analyses were performed to ensure there was no violation of the assumption of normality, linearity, and multicollinearity. A significant regression equation as found (F (3,35) = 4.853, *p* = 0.006 with adjusted R square of 0.597). IL-18 predicted level is equal to 40.531+ 0.648 (TJ) + 0.247 (TG) + 0.105 (IL-17). IL-18 level increased by 0.648 for every tender joint, by 0.247 for every 1 mg/dL of TG, and by 0.105 for every pg/mL of IL-17. TJ, TG, and IL-17 were significant predictors (Table 4).

Similarly, analyzing further covariates for TG in linear regression model based on AICc and wi indices was calculated for patients with perPsA. We have found that TG level (the dependent variable) was influenced mainly by IL-18 levels and secondly by BMI (Table 4). Multiple linear regression was calculated to predict TG concentration based on IL-18 concentration, BMI, and CRP level. Preliminary analyses were performed to ensure there was no violation of the assumption of normality, linearity, and multicollinearity. A significant regression equation as found (F (3,47) =7.533, *p* < 0.001 with adjusted R square of 0.3. TG predicted level is equal to −42.709+ 0.45 (BMI) + 0.45 (IL-18) − 0.1 (CRP)). TG level increased by 0.45 for every kg per square meter, by 0.45 for every pg/mL of IL-18, and decreased by 0.10 for every 1 mg/L of CRP. IL-18 and BMI were significant predictors (Table 4).

## 4. Discussion

In our study, we examined IL-18 as a potential mediator of disease activity and an additional factor associated with an increased cardiovascular risk in PsA. AS and PsA are both classified as autoinflammatory spondyloarthritis, yet patients with PsA seem to have more “traditional” cardiovascular risk factors [28] and a higher risk for acute coronary syndrome and stroke [29]. Interestingly, our study showed that those differences in the lipid profile and cardiovascular risk between AS and PsA might be related to IL-18. We have found that patients with PsA, especially with peripheral arthritis, presented more proatherogenic lipid profile (higher triglycerides concentration, lower HDL concentration), more proatherogenic biometric profile (common occurrence of obesity) than AS patients (Table 1). Our findings also indicate significant differences in lipid and biometric profiles between patients with perPsA and axPsA. However, compared with the observations of Puche-Larrubia et al. [30], the number of patients in our study was much smaller, which did not allow for the correct adjustment of the groups in relation to the duration of the disease, gender, and other factors. Undoubtedly, the information indicating differences in lipid and biometric profiles between the groups and higher CV risk in perPsA requires further research. Patients with peripheral arthritis revealed lower HDL and higher TG concentrations (Table 1) and a stronger tendency towards obesity (Table 2) than patients with AS and axial PsA, which implies the existence of an increased cardiovascular risk in the perPsA group. Age, sex, and the duration and activity of the disease in the groups we compared did not differ significantly (Table 1). So far, several studies have investigated factors suspected of being associated with an increased cardiovascular risk and with disease activity in patients with PsA. Among the most indicated factors were ESR, CRP [29,31], proinflammatory cytokines (TNF, IL-17, IL-1β) [32,33,34], and particular clinical symptoms such as peripheral arthritis and dactylitis [31,35]. Interestingly, none of these factors seemed to be connected directly to a highly proatherogenic metabolic profile in PsA. In our previous research, we observed higher serum IL-18 and IL-17 concentrations in SpA patients than in healthy controls, suggesting an association of these cytokines with increased disease activity [36,37]. Our current investigation revealed that PsA patients had significantly higher serum levels of IL-18 compared with the AS group but similar levels of IL-17 (Table 1). In addition, a significant correlation between IL-18 and IL-17 was found in the PsA group. These observations may indicate that both cytokines, IL-18 and IL-17, contribute to the development of inflammation in PsA, but possibly not in AS, where the action of IL-17 is widely recognized as dominant [15,16]. 

Moreover, our data show that altered lipid profile in PsA is associated with higher serum concentrations of IL-18 (Figure 1, Figure 2, Figure 3). Although we do not have direct evidence of the effect of IL-18 on alterations in blood lipid levels, we have managed to find that IL-18 levels correlate significantly and positively with triglycerides concentrations and AI value, but inversely with HDL levels, especially in patients with perPsA arthritis (compare Figure 1 and Figure 3). In perPsA patients, alterations to the lipid profile were stronger than in patients with an axial form of PsA, which seems to be associated with the higher levels of IL-18 and, in addition, with the number of inflamed joints. Interestingly, the regression models show that cytokine (IL-18, IL-17) concentrations together with hypertriglyceridemia and tender joint counts in perPsA are connected (Table 3 and Table 4). According to our observations, obesity and hypertriglyceridemia in PsA are intertwined with disease activity via IL-18 action. On the contrary, similar interactions between metabolic profile and IL-18 or IL-17 were not observed in AS.

Interesting findings were obtained in a small group (*n* = 14) of patients with PsA and concomitant ischemic heart disease (PsA + IHD). In this particular group, we found a stronger positive correlation between IL-18 levels and concentrations of triglycerides and AI than in all PsA patients (compare Figure 1 and Figure 2). We also observed a much stronger negative correlation between IL-18 and HDL concentrations in PsA patients with IHD (Figure 2A–C). Moreover, we have found a strong positive correlation between disease activity of PsA (measured by DAPSA) and IL-18 concentrations (Figure 2D).

All these data strengthen the hypothesis that there is a connection between IL-18, systemic inflammation, dyslipidemia, and atherosclerosis in PsA. However, the exact mechanism of action of IL-18 in these areas remains unexplained. There are some data from ex vivo investigations and a knockout mice study showing that both IL-18 and IL-17 are involved in the development of inflammation, suggesting the synergistic role of IL-18 and IL-17 in the development of atherosclerosis [38,39,40]. Inspiring connections regarding IL-18 and inflammation were found in recently discovered IL-18 receptor-related autoinflammatory disease, where systemic inflammation is directly mediated by the IL-18 pathway [41,42]. Co-occurrence of similar associations between IL-18-driven inflammation and altered lipid metabolism can be found in several studies on other autoinflammatory and metabolic conditions [41,42]. Weiss et al. reported that activity of macrophage activation syndrome (MAS) was associated with elevated IL-18 levels [43], while Ravelli et al. established triglycerides level exceeding 160mg/dl as one of the major diagnostic criteria for MAS [41]. Inflammatory rheumatic diseases, occurring in both adults and children, are associated with various types of disorders in the cardiovascular system. For example, adult patients with juvenile idiopathic arthritis (JIA) often present subclinical signs of endothelial damage and thus increased CV risk [44]. Our current research suggests the need for further studies on the role of IL-18 in the pathogenesis of vascular injury in rheumatic diseases, including JIA.

Moreover, similar associations were found in studies on the impact of postprandial hyperglycemia and hypertriglyceridemia on endothelial inflammation [45,46,47]. IL-18 may attenuate lipid metabolism, postprandial hyperglycemia, and modulate insulin signalling [45] and by triggering endothelial inflammation, promote the formation of atherosclerotic plaques [45,46,47]. Finally, it is also worth noting that after IL-1β blockade with canakinumab in patients with ischemic heart disease, in the CANTOS trial, IL-18 remained as the main proinflammatory factor associated with increased CV risk [48]. All these findings support our observation indicating the unique role of IL-18 in the pathways of systemic inflammation and formation of proatherogenic metabolic profile in PsA, which is a novel observation in SpA.

Our study adds new information, but, unfortunately, it is burdened with flaws typical for this kind of research. First, our methodology did not include prospective observation, and therefore our outcomes are based on statistical relationships between investigated factors, clinical records, etc. We included a relatively low number of patients for CV risk research (*n* = 154), and the number of patients with diagnosed ischemic heart disease was small indeed. There is no doubt that observations made in this group require further, more extended studies.

## 5. Conclusions

We report that patients with perPsA reveal a more pronounced proinflammatory and cardiovascular risk profile than patients with axPsA and AS. Importantly, in PsA, but not in AS, we have found that IL-18 is associated with higher disease activity and proatherogenic lipid profile, leading to a higher cardiovascular risk. Our present results inspire further research of the role of IL 18 in the complex pathomechanism(s) of PsA.

## Figures and Tables

**Figure 1 jcm-11-00766-f001:**
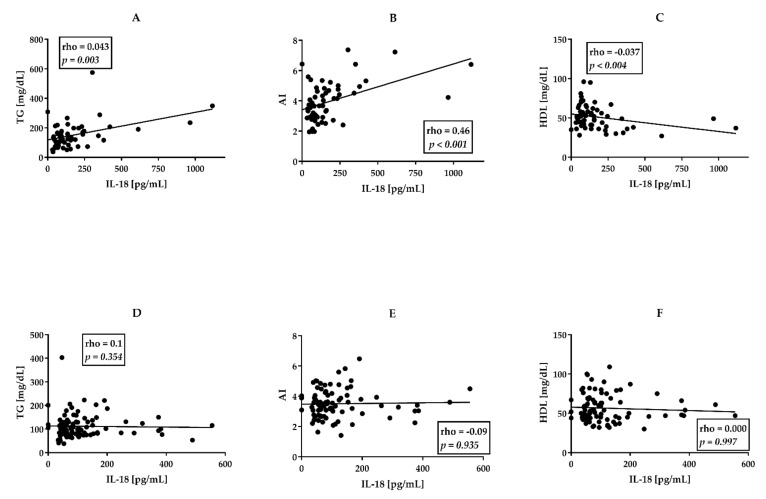
Correlations between IL-18 and lipid profile in patients with PsA ((**A**–**C**), *n* = 61) in comparison with AS patients ((**D**–**F**), *n* = 94). AI—atherogenic index. TG—triglycerides. HDL—High-Density Cholesterol. All correlations were analyzed using Spearman’s rank test. Rho—Spearman’s rank correlation coefficient. *p*—value for null hypothesis for Spearman’s rank correlation.

**Figure 2 jcm-11-00766-f002:**
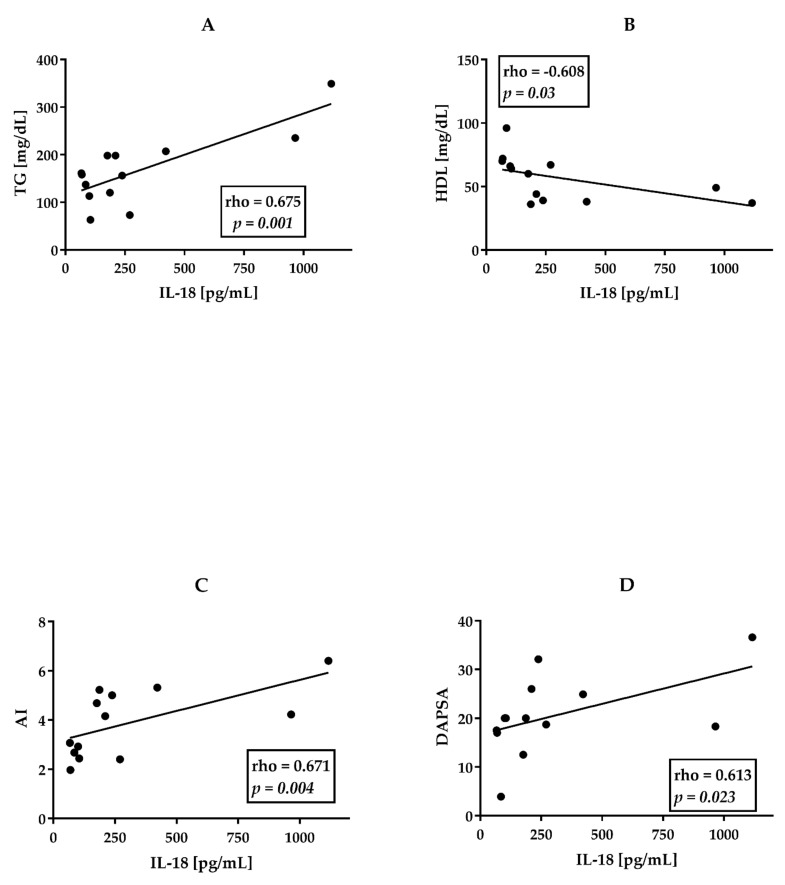
Correlations between IL-18 and lipid profile and DAPSA in patients with PsA and ischemic heart disease (*n* = 14). (**A**)TG—triglycerides. (**B**) HDL—High-Density Cholesterol. (**C**) AI—atherogenic index. (**D**) DAPSA. All correlations were analyzed using Spearman’s rank test. Rho—Spearman’s rank correlation coefficient. *p*—value for null hypothesis for Spearman’s rank correlation.

**Figure 3 jcm-11-00766-f003:**
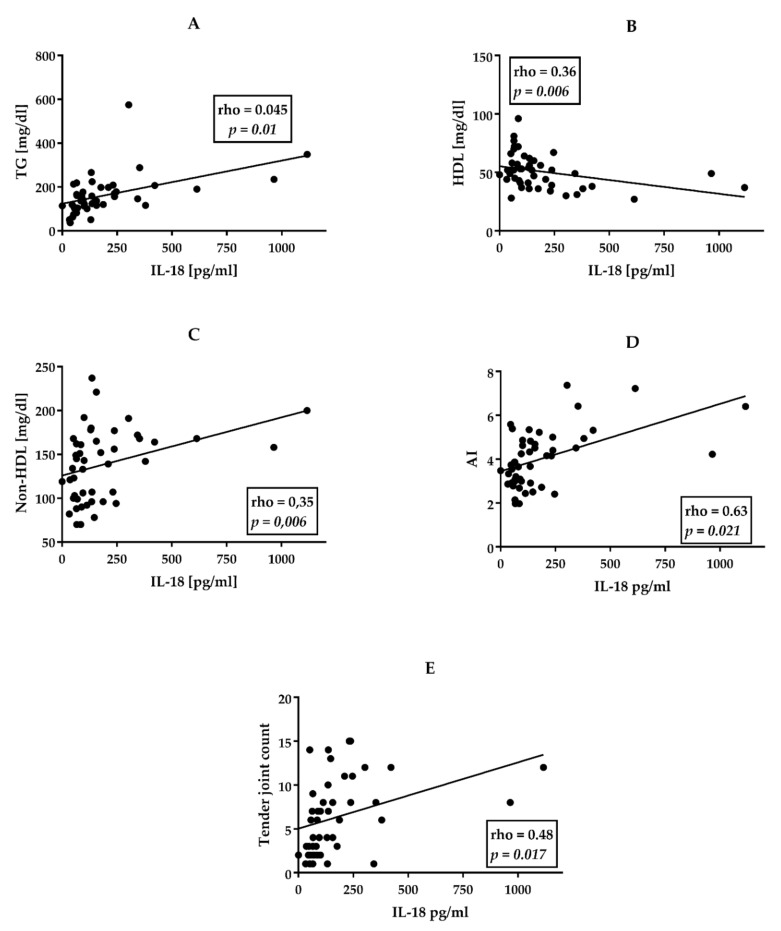
Correlations between IL-18 and lipid profile and TJ in patients with perPsA (*n* = 47). (**A**) TG- triglycerides. (**B**) HDL—High-Density Cholesterol. (**C**) Non-HDL-C—Non-High-Density Cholesterol. (**D**) AI—atherogenic index. (**E**) TJ—tender joint count. All correlations were analyzed using Spearman’s rank test. Rho—Spearman’s rank correlation coefficient. *p*—value for null hypothesis for Spearman’s rank correlation.

**Table 1 jcm-11-00766-t001:** Demographic and clinical characteristics of AS and PsA patients.

Measured Parameters	AS *n* = 94	PsAtotal *n* = 61	axPsA *n* = 14	perPsA *n*= 47	Normal Ranges	*p K–W*
Age [years]	43.5 (IQR 18)	46.5 (IQR 19)	43 (IQR 19)	47 (IQR 18.8)		ns
male	67.7%	52.5%	71.4%	48.9%		ns
female	32.3%	47.5%	29.6%	51.1%		ns
Disease duration time [years]	5 (IQR 10)	4.5 (IQR 8.3)	4 (IQR 10)	4.5 (IQR 9)		ns
		Disease activity				
BASDAI	5.8 (IQR 3.1)		5.8 (IQR 4.1)			ns
ASDAS CRP	3.6 (IQR 2.8)		3.1 (IQR 2.3)			0.011
DAPSA		21.3 (IQR 16.8)	18.7 (IQR 39)	28 (IQR 21.15)		
		Biochemical parameters				
CRP [mg/L]	8.5 (IQR 13)	9 (IQR 12)	12.4 (IQR 12)	6 (IQR 13)	1–5	ns
ESR. [mm/h]	16 (IQR 22)	11 (IQR 21)	14 (IQR 13.5)	12 (IQR 22)	1–10	ns
Fasting glucose [mg/dL]	89.5 (IQR 9)	94 (IQR 15)	94 (IQR 16)	93.5 (16.3)	60–100	0.02
Atherogenic index. TC/HDL-AI	3.45 (1.2)	3.7 (IQR 1.7)	3.2 (IQR 3.14)	3.8 ( IQR 1.6)	4.5/4.0 3.5/3	0.024
Total Cholesterol [mg/dL]	185 (IQR 51)	183 (IQR 62)	165 (IQR 60)	183 (IQR 62)	190	ns
LDL [mg/dL]	108.5 (IQR 38.8)	105.8 (IQR 50.7)	94.2(IQR 50.7)	110.2 (IQR 51.4)	110/90/70/50 *	ns
Non-HDL cholesterol [mg/dL]	127 (IQR 47)	138 (IQR 53.5)	111 (IQR 64)	145 (IQR 52.7)	145/130/100/100 *	ns
HDL [mg/dL]	55.5 (IQR 25)	49 (IQR 17)	52 (IQR 45)	48 (IQR 19)	40–50	0.025
TG [mg/dL]	98.5 (IQR 52)	137 (IQR 77)	121 (IQR 95)	160 (IQR 91)	35–150	<0.001
Uric Acid [mg/dL]	5.1 (IQR 1.8)	5.5 (IQR 2)	4.6 (IQR 2.8)	5.3 (IQR 1.6)	4–5	ns
		Biometric indices				
BMI [kg/m2]	25.5 (IQR 6)	29 (IQR 7.6)	26 (IQR 5.3)	29 (IQR 8.2)	18.5–24.9	0.02
Waist circumference [cm]	90 (IQR 15)	94 (IQR 22)	94 (IQR 15.8)	94 (IQR 16)		0.044
Hip circumference [cm]	95 (IQR 13)	101 (IQR 17)	95 (IQR 14)	102 (IQR 19)		0.007
		Cytokine profile				
IL-18 [pg/mL]	80 (IQR 68)	140 (IQR 161.9)	118( IQR165)	160 (IQR 137)		<0.001
IL 17 [pg/mL]	1 (IQR 0.7)	1.3 (IQR 1.3)	1 (IQR 0.6)	1.13 (IQR 0.8)		ns
		Treatment				*p* Chi^2^
Methotrexate (15–25 mg per week)	17 (18%)	38 (62.3%)	8 (57%)	30 (63.8%)		<0.001
Sulphasalazine (2–3 grams/day)	20 (21.3%)	21 (34.4%)	3 (21.4%)	18 (38.3%)		ns
Cyclosporine (3–5 mg/kg per day)	0	5 (8.5%)	0	5 (10.6%)		ns
Steroids (the equivalent of 5–15 mg prednisone per day)	6 (6.51%)	5 (8.5%)	2(14.3%)	3 (6.4%)		ns
NSAIDS constantly	69 (73.4%)	23 (37.7%)	7 (50%)	16 (34%)		<0.001
NSAIDS “on demand”	7 (7.54%)	17 (27.9%)	8 (57%)	9 (19%)		ns
Physical therapy	23 (24.5%)	11 (18%)	2 (14.3%)	9 (19%)		ns

Differences statistically significant with *p* < 0.05 are marked. Data are expressed as median with IQR. AS—ankylosing spondylitis, axPsA—PsA patients with the axial disease. perPsA—PsA patients with peripheral arthritis patients with peripheral joint inflammation. *—divided based on ESC guidelines by cardiovascular risk groups as follows: low, moderate, high, very high/extreme. Data regarding patients treatment were analyzed with Chi^2^ test rest of the data were analyzed using Kruskal–Wallis test. *p* K–W—value for null hypothesis for Kruskal–Wallis test. *p* Chi^2^—value for null hypothesis for Chi^2^ test.

**Table 2 jcm-11-00766-t002:** Cardiovascular risk factors in axial PsA, peripheral PsA and AS groups. Differences statistically significant with *p* < 0.05 are marked.

Cardiovascular Risk Factors	AS *n* = 94	Axial PsA *n* = 14	Peripheral PsA *n* = 47	*p* *chi^2^*		Post Hoc Tests	
					*p* (AS vs. axPsA)	*p* (AS vs. perPsA)	*p* (axPsA vs. perPsA)
Smoking more than 1 cigarette per day for 30 days	43 (45.7%)	3 (21.4%)	21 (44.7%)	ns	ns	ns	ns
Physical activity more than 120 min twice weekly	21 (22.3%)	2 (14.3%)	11 (23.4%)	ns	ns	ns	ns
Hyperuricemia	13 (13.8%)	5 (35.7%)	3 (6.4%)	ns	ns	ns	ns
Hypertriglyceridemia	15 (16%)	2 (14.3%)	23 (48.9%)	<0.001	<0.001	<0.001	<0.001
Obesity BMI > 30kg/m^2^	16 (17%)	3 (21.4%)	20 (42.6%)	ns	ns	ns	ns
Dyslipidemia in patients older than 40 years old	17 (18.1%)	3 (21.4%)	7 (14.9%)	ns	ns	ns	ns
Dyslipidemia in patients younger than 40 years old	24 (25.5%)	4 (28.6%)	12 (25.5%)	ns	ns	ns	ns
Hypertension	30 (32%)	5 (35.7%)	21 (44.78%)	ns	ns	ns	ns
Ischaemic heart disease	9 (9.6%)	0 (0.00%)	14 (29.8%)	0.002	0.05	0.05	0.049

*p* chi^2^—value for null hypothesis for Chi^2^ test. AS—ankylosing spondylitis, axPsA—PsA patients with the axial disease. perPsA—PsA patients with peripheral joint inflammation.

**Table 3 jcm-11-00766-t003:** Linear regression model for IL-18 in perPsA group (*n* = 47).

Model	AICc	Delta AICc	Wi	Adjusted R2
Tender joint count	596.376	0	0.01	0.593
Tender joint count + TG	589.325	7.051	0.32	0.566
Tender joint count + TG + IL-17	587.842	8.534	0.67	0.597
Coefficient of determination from the regression with confidence intervals				
Coefficient	B	CI	*p*	
Tender joint count	0.648	269.12 (130:408)	<0.001	
TG	0.247	0.946 (0.299:1.592)	<0.001	
IL-17	0.105	3.193 (−0.151:6.537)	0.06	

The coefficient of determination—R2. Sampling weights—Wi. Akaike information criterion—AICc. Confidence interval—CI. B—beta coefficient.

**Table 4 jcm-11-00766-t004:** Linear regression model for TG levels in perPsA. group (*n* = 47).

Model	AICc	Delta AICc	Wi	Adjusted R2
IL-18	413.589	0	0.88	0.17
IL-18 + BMI	408.146	5.443	0.06	0.299
IL-18 + BMI + CRP	408.091	5.498	0.06	0.3
Coefficient of determination from the regression with confidence intervals				
Coefficient	B	CI	*p*	
BMI	0.45	8.24 (3.03:13.451)	<0.001	
IL-18	0.45	0.314 (0.116: 0.511)	<0.001	
CRP	−0.1	−1.12 (−2.36: 0.334)	0.109	

The coefficient of determination—R2 Sampling weights–Wi. Akaike information criterion—AICc. Confidence interval—CI. B—beta coefficient.

## Data Availability

Data are available upon reasonable request according to ICMJE requirements: Data shared: All the individual participant data collected during the trial, after deidentification. Available documents: study protocol, statistical analysis, analytic code. When will data be available: beginning three months and ending two years following article publication. With whom: Researchers who provide a methodologically sound proposal. For what type of analyses: to achieve the aim of the approved proposal. By what mechanism will data be made available: The proposal should be directed to Krzysztof.bonek@spartanska.pl.

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
