# Peer review of "Associations of IL-18 with Altered Cardiovascular Risk Profile in Psoriatic Arthritis and Ankylosing Spondylitis"

_jcm, 2022, doi:10.3390/jcm11030766_

Round 1
Reviewer 1 Report
This investigation is additional evidence that pediatric rheumatology and cardiology have contiunous connection during development. The mayority of rhematic diseases (FR, SLE, JDM, SjS, Moy-Moya and other vasculitides..... and PsA.) have the simptom from cardiac side; aortic and mitraln valve, endocarditis, cardiomyopathy, CCAVC during pregnancy, obstructive lesions of different vessels etc. +IHC. I reccomend optional entern this observation into the manuscrit.
Author Response
Dear Reviewer,
Thank you for your review. We have revised the manuscript according to your suggestions. Changes to the manuscript were made as follows:
This investigation is additional evidence that pediatric rheumatology and cardiology have contiunous connection during development. The mayority of rhematic diseases (FR, SLE, JDM, SjS, Moy-Moya and other vasculitides..... and PsA.) have the simptom from cardiac side; aortic and mitraln valve, endocarditis, cardiomyopathy, CCAVC during pregnancy, obstructive lesions of different vessels etc. +IHC. I reccomend optional entern this observation into the manuscrit.
In line 310-314 following lines were added:
Inflammatory rheumatic diseases, occurring in both adults and children, are associated with various types of disorders in the cardiovascular system. For example, adult patients with juvenile idiopathic arthritis (JIA) often present subclinical signs of endothelial damage and thus increased CV risk (Ref 2). Our current research suggests the need for further studies on the role of IL-18 in the pathogenesis of vascular injury in various rheumatic diseases including JIA.

Reviewer 2 Report
Comments: Associations of IL-18 with altered cardiovascular risk profile in psoriatic arthritis and ankylosing spondylitis
Bonek et al. presented the study on the associations of IL-18 with altered cardiovascular (CV) risk profile in psoriatic arthritis (PsA) and ankylosing spondylitis (AS). The association between chronic inflammatory arthritis, such as rheumatoid arthritis, PsA, and AS, and CV risk is well established[1]. However, much of the knowledge about CV comorbidities in rheumatic patients, its epidemiology, and its underlying mechanisms comes from studies in patients with RA, while such data are limited in PsA. In this study, the authors insisted that in PsA, especially peripheral PsA (perPsA), elevated serum IL-18 level is associated with higher disease activity and pro-atherogenic lipid profile, especially triglyceride (TG), leading to a higher CV risk. However, some issues to be clarified before this manuscript is further considered.
Major concerns;
- CV risk depending on the subtype of PsA
The authors described that ischemic heart disease (IHD) was detected only in patients with perPsA among the patients with PsA. They insisted that “our findings indicated significant differences in lipid and biometric profiles between patients with perPsA and axPsA, which is quite new information (Reference 46. Puche-Larrubia MA, Ther Adv Musculoskelet Dis. 2021 Sep 20;13:1759720X211045263.)” in lines 218-219 on page 10 in ‘Discussion’ section. However, Puche-Larrubia et al. showed that “axial involvement with psoriasis patients exhibited a lower prevalence of dyslipidemia than peripheral involvement with psoriasis, but differences in dyslipidemia disappeared after adjusting for disease duration, sex and country. No differences were found between these two groups in CV diseases (i.e. IHD and stroke).” Furthermore, the number of axPsA was small (only 14 patients) in this study. Therefore, further evaluation and discussion of CV risk depending on the subtype of PsA may be needed.
2. Concerns about data clarity
1) In Table 1, although the unit of CRP is described as mg/dl and normal value of CRP is not shown, considering that the median levels of CRP are 8.5 and 9 in AS and PsA patients, respectively, and those of ESR are not so high (median level, 16 and 11 mm/hr, respectively), the unit of CRP is assumed to be mg/L. Assuming that the unit of CRP is mg/L, the enrolled patients demonstrated high or very high disease activity based on their values of BASDAI, ASDAS-CRP, and DAPSA, compared to the levels of ESR and CRP[2-4].
2) The same data is described differently for each table. For example, atherogenic index (AI) in AS group is 3.46 ± 1 in Table 1 but 3.45 ± 1.01 in Table 3. In addition, total cholesterol (TC) in AS group is 186.82 ± 38.89 in Table 1 but 187.07 ± 39.03 in Table 3.
3) In Table 2, IHD was observed in 27.08% of patients with perPsA. As shown in Figure 2 and Table 2, the number of PsA patients with IHD was 14 and none of the axPsA patients had IHD. Calculated using the data in Figure 2, the prevalence of IHD in patients with perPsA was 29.78% (14/47).
4) The authors did not show the clinical data, such as enthesitis, dactylitis, and unveitis, in Table 1. In addition, they did not demonstrate any correlation between serum IL-18 level and clinical data, treatment modality, or acute phase reactants in this manuscript. Nevertheless, they described that clinical data, treatment modality, or acute phase reactants were not associated with the serum IL-18 level in the linear regression model for IL-18 in lines 192-196 on page 8 in “Result” section.
- Concerns about statistical analysis
1) Linear regression model
- The format of Table 4 & 5, which shows the results of linear regression models, is not usual and there’s no footnote in the bottoms of tables, so it is hard to interpret the contents of the tables.
- The authors described that “IL-17, TG concentration, and tender joint count (TJC) are associated with serum IL-18 level” in lines 195-196 on page 8 in ‘Result’ section. The authors selected the linear regression model with variables of “TJC + TG + IL-17” and further analyzed the association between each variable and serum IL-18 level. However, compared the model with variables of “TJC + TG + IL-17” to that with variables of “TJC + TG”, the values of delta AICc of two models were not so different (8.534 vs. 7.051). This suggested that adding the variable of IL-17 in the linear model for IL-18 did not significantly affect the statistical results.
- In addition, while the correlation between DAPSA and serum IL-18 level was shown (although presented only in PsA patients with IHD), the correlation between tender joint count and serum IL-18 level was not demonstrated in this manuscript.
2) Statistical significance
- In Table 2, the p value of obesity (BMI>30) between AS and axPsA groups was 0.09 (AS vs. axPsA,17.44% vs.18.18%). On the other hand, the p value of obesity between AS vs. perPsA groups was 0.009 (AS vs. perPsA ,17.44% vs. 41.86%).
- In Table 3, the p value of HDL between AS and axPsA groups was 0.05 (AS vs. axPsA, 56.84 ± 17.68 vs.49.92 ± 9.36). On the other hand, the p value of HDL between axPsA and perPsA groups was 0.025 (axPsA vs. perPsA ,49.92 ± 9.36 vs.49.55 ± 17.19).
Minor concerns;
- Please add footnotes, which explain the abbreviation and statistical analysis, in the bottom of each table and figure.
- Please state consistently below the decimal point of the data in the tables.
- Please make data descriptions consistent. For example, in Table 1, HDL levels are expressed as the median values with IQR, but in Table 3, the same HDL levels are described as the mean values ± SD.
- Some of the contents of Table 1 and contents of Table 3 overlap, so it would be better to combine these tables.
- Please specify normal values of each parameter in Table 1.
- Please specify the number of patients of each parameter in Table 2.
- Please write down the references according to the guideline.
References
- Eder, L.; Harvey, P. Cardiovascular Morbidity in Psoriatic Arthritis: What Is the Effect of Inflammation? J Rheumatol 2017,44(9), 1295-1297.
- Ramiro, S.; van der Heijde, D.; van Tubergen, A.; Stolwijk, C.; Dougados, M.; van den Bosch, F.; Landewé, R. Higher disease activity leads to more structural damage in the spine in ankylosing spondylitis: 12-year longitudinal data from the OASIS cohort. Ann Rheum Dis 2014, 73, 1455-1461.
- Machado, P.M.; Landewe, R.; Heijde, D.V. Assessment of SpondyloArthritis international S. Ankylosing Spondylitis Disease Activity Score (ASDAS): 2018 update of the nomenclature for disease activity states. Ann Rheum Dis 2018, 77, 1539-1540.
Schoels M, Aletaha D, Funovits J, Kavanaugh A, Baker D, Smolen JS. Application of the DAREA/DAPSA score for assessment of disease activity in psoriatic arthritis. Ann Rheum Dis. 2010; 69(8):1441-7.
- Schoels, M.M.; Aletaha, D.; Alasti, F.; Smolen, J.S. Disease activity in psoriatic arthritis (PsA): defining remission and treatment success using the DAPSA score. Ann Rheum Dis 2016, 75(5), 811-818.
Author Response
Dear Reviewer,
Thank you for your review. We have revised the manuscript according to your suggestions. Changes to the manuscript were made as follows:
Major concernsCV risk depending on the subtype of PsA
The authors described that ischemic heart disease (IHD) was detected only in patients with perPsA among the patients with PsA. They insisted that “our findings indicated significant differences in lipid and biometric profiles between patients with perPsA and axPsA, which is quite new information (Reference 46. Puche-Larrubia MA, Ther Adv Musculoskelet Dis. 2021 Sep 20;13:1759720X211045263.)” in lines 218-219 on page 10 in ‘Discussion’ section. However, Puche-Larrubia et al. showed that “axial involvement with psoriasis patients exhibited a lower prevalence of dyslipidemia than peripheral involvement with psoriasis, but differences in dyslipidemia disappeared after adjusting for disease duration, sex and country. No differences were found between these two groups in CV diseases (i.e. IHD and stroke).” Furthermore, the number of axPsA was small (only 14 patients) in this study. Therefore, further evaluation and discussion of CV risk depending on the subtype of PsA may be needed.
In accordance with the comments of the Reviewer, the following changes were made in lines 254-260:
“Our findings also indicate significant differences in lipid and biometric profiles between patients with perPsA and axPsA. However, compared to the observations of Puche-Larrubia et al. (46), the number of patients in our study was much smaller, which did not allow for the correct adjustment of the groups in relation to the duration of the disease, gender and other factors. Undoubtedly, the information indicating differences in lipid and biometric profiles between the groups and higher CV risk in perPsA requires further research”
2. Concerns about data clarity
1)In Table 1, although the unit of CRP is described as mg/dl and normal value of CRP is not shown, considering that the median levels of CRP are 8.5 and 9 in AS and PsA patients, respectively, and those of ESR are not so high (median level, 16 and 11 mm/hr, respectively), the unit of CRP is assumed to be mg/L. Assuming that the unit of CRP is mg/L, the enrolled patients demonstrated high or very high disease activity based on their values of BASDAI, ASDAS-CRP, and DAPSA, compared to the levels of ESR and CRP[2-4].
Based on current research it has been reported that despite low or normal values of CRP in AS and PsA patient’s disease is considered as active. Recent studies, such as Sokolova et al. [1], presented that similarly to our study, inflammatory mediators expression pattern is mostly related with disease phenotype. Also, in study by Brown et al. 110 out of 197 patients with moderate to severe AS enrolled into MEASURE 1 and MEASURE 2 studies had lower CRP than 10mg/l [2].
[1] Sokolova, M.V., Simon, D., Nas, K. et al. A set of serum markers detecting systemic inflammation in psoriatic skin, entheseal, and joint disease in the absence of C-reactive protein and its link to clinical disease manifestations. Arthritis Res Ther 22, 26 (2020). https://doi.org/10.1186/s13075-020-2111-8
[2] Braun J, Deodhar A, Landewé R, et al Impact of baseline C-reactive protein levels on the response to secukinumab in ankylosing spondylitis: 3-year pooled data from two phase III studies RMD Open 2018;4:e000749. doi: 10.1136/rmdopen-2018-000749
2) The same data is described differently for each table. For example, atherogenic index (AI) in AS group is 3.46 ± 1 in Table 1 but 3.45 ± 1.01 in Table 3. In addition, total cholesterol (TC) in AS group is 186.82 ± 38.89 in Table 1 but 187.07 ± 39.03 in Table 3.
Thank you for pointing out these errors - according to reviewer’s suggestions the manner of data presentation has been unified. All data in tables and in the text are shown as medians with IQR.
3) In Table 2, IHD was observed in 27.08% of patients with perPsA. As shown in Figure 2 and Table 2, the number of PsA patients with IHD was 14 and none of the axPsA patients had IHD. Calculated using the data in Figure 2, the prevalence of IHD in patients with perPsA was 29.78% (14/47)
All data were checked and unified in tables and in the text.
4) The authors did not show the clinical data, such as enthesitis, dactylitis, and unveitis, in Table 1. In addition, they did not demonstrate any correlation between serum IL-18 level and clinical data, treatment modality, or acute phase reactants in this manuscript. Nevertheless, they described that clinical data, treatment modality, or acute phase reactants were not associated with the serum IL-18 level in the linear regression model for IL-18 in lines 192-196 on page 8 in “Result” section.
Additional Table for supplementary material (supplementary Table S1) with frequency of enthesitis, dactylitis, uveitis and occurrence of HLA-B27 was added (lines 364-371)
3. Concerns about statistical analysis
1) Linear regression model
- The format of Table 4 & 5, which shows the results of linear regression models, is not usual and there’s no footnote in the bottoms of tables, so it is hard to interpret the contents of the tables.
We have changed format of table 4 and 5 to make data presentation clear. Also, we have added standard data including regression equation, standardized B, degrees of freedom according to AMA standard in lines 214-220 and 229-236.
- The authors described that “IL-17, TG concentration, and tender joint count (TJC) are associated with serum IL-18 level” in lines 195-196 on page 8 in ‘Result’ section. The authors selected the linear regression model with variables of “TJC + TG + IL-17” and further analyzed the association between each variable and serum IL-18 level. However, compared the model with variables of “TJC + TG + IL-17” to that with variables of “TJC + TG”, the values of delta AICc of two models were not so different (8.534 vs. 7.051). This suggested that adding the variable of IL-17 in the linear model for IL-18 did not significantly affect the statistical results.
Justification for further analysis of models was based in Aikake weights (wi) due to its superiority over AICc alone [1] and has been added with reference in lines 117-122. According to reviewer’s observation minor change in delta AICc was noted and referred to in lines 211-214
[1] Wagenmakers EJ, Farrell S. AIC model selection using Akaike weights. Psychon Bull Rev. 2004 Feb;11(1):192-6. doi: 10.3758/bf03206482. PMID: 15117008.
- In addition, while the correlation between DAPSA and serum IL-18 level was shown (although presented only in PsA patients with IHD), the correlation between tender joint count and serum IL-18 level was not demonstrated in this manuscript.
Correlation between tender joints count and IL-18 has been added to Figure 3 as Fig. 3 E
2) Statistical significance
- In Table 2, the p value of obesity (BMI>30) between AS and axPsA groups was 0.09 (AS vs. axPsA,17.44% vs.18.18%). On the other hand, the p value of obesity between AS vs. perPsA groups was 0.009 (AS vs. perPsA ,17.44% vs. 41.86%).
As reviewer has noted non-significant data should not be presented therefore presentation of the data was changed. Also in Table 3 chi-square test was with post hoc testing for intergroup comparison with alpha level of 0.05. All data were checked once again and presented as number of patients and percentages.
- In Table 3, the p value of HDL between AS and axPsA groups was 0.05 (AS vs. axPsA, 56.84 ± 17.68 vs.49.92 ± 9.36). On the other hand, the p value of HDL between axPsA and perPsA groups was 0.025 (axPsA vs. perPsA ,49.92 ± 9.36 vs.49.55 ± 17.19).
Tables 1 and 3 have been merged into Table 1 according to reviewer suggestion. Presentation of data based on means instead of medians, lead to misleading information. Due to changing of all data into medians it currently is presented as HDL AS 55.5 vs axPsA 52 vs perPsA 48 mg/dl. Differences in p-values seem to be originating from HDL-values dispersion (IQR AS 25 vs axPsA 45 vs perPsA 19).
Minor concerns;
3. Please add footnotes, which explain the abbreviation and statistical analysis, in the bottom of each table and figure.
Footnotes explaining abbreviation were added under figures and tables.
4. Please state consistently below the decimal point of the data in the tables
The manuscript has been reformatted to match reviewers’ concerns
5. Please make data descriptions consistent. For example, in Table 1, HDL levels are expressed as the median values with IQR, but in Table 3, the same HDL levels are described as the mean values ± SD.
All data have been presented as median with IQR (in tables and in the text)
6. Some of the contents of Table 1 and contents of Table 3 overlap, so it would be better to combine these tables.
Tables 1 and 3 were merged into Table 1
7. Please specify normal values of each parameter in Table 1.
This issue has already been answered in Major Concerns section 1
8. Please specify the number of patients of each parameter in Table 2.
Number of patients with percentages was added in Table 2
9. Please write down the references according to the guideline.
References were reformatted according to reviewer’s guideline with ENDNOTE

Round 2
Reviewer 2 Report
Thank you for the efforts of the revision.
I suggest a few minor revisions in this manuscript.
I. Major concerns; CV risk depending on the subtype of PsA
- Concerns about data clarity
- In Table 1, although the unit of CRP is described as mg/dl and normal value of CRP is not shown, considering that the median levels of CRP are 8.5 and 9 in AS and PsA patients, respectively, and those of ESR are not so high (median level, 16 and 11 mm/hr, respectively), the unit of CRP is assumed to be mg/L. Assuming that the unit of CRP is mg/L, the enrolled patients demonstrated high or very high disease activity based on their values of BASDAI, ASDAS-CRP, and DAPSA, compared to the levels of ESR and CRP[2-4].
Authors' answer: Based on current research it has been reported that despite low or normal values of CRP in AS and PsA patient’s disease is considered as active. Recent studies, such as Sokolova et al. [1], presented that similarly to our study, inflammatory mediators expression pattern is mostly related with disease phenotype. Also, in study by Brown et al. 110 out of 197 patients with moderate to severe AS enrolled into MEASURE 1 and MEASURE 2 studies had lower CRP than 10mg/l [2].
-> I suggest putting this paragraph in “Results” section.
II. Minor concerns;
3. Please add footnotes, which explain the abbreviation and statistical analysis, in the bottom of each table and figure.
Authors' answer: Footnotes explaining abbreviation were added under figures and tables.
- The footnotes of all pictures and tables are not properly modified. Footnote should have statistical description (such as Pearson's or Spearman's correlations, etc.) and abbreviation to facilitate interpretation of contents in pictures or tables. In particular, footnote of Table 1 does not describe the meaning of p values.
4. Please state consistently below the decimal point of the data in the tables
Authors' answer: The manuscript has been reformatted to match reviewers’ concerns
- Unmodified descriptions are observed in Table 1. Please revise them further.
Author Response
Thank you for the review. We have modified manuscript, following Your revisions.
- Major concerns; CV risk depending on the subtype of PsA
- Concerns about data clarity
- In Table 1, although the unit of CRP is described as mg/dl and normal value of CRP is not shown, considering that the median levels of CRP are 8.5 and 9 in AS and PsA patients, respectively, and those of ESR are not so high (median level, 16 and 11 mm/hr, respectively), the unit of CRP is assumed to be mg/L. Assuming that the unit of CRP is mg/L, the enrolled patients demonstrated high or very high disease activity based on their values of BASDAI, ASDAS-CRP, and DAPSA, compared to the levels of ESR and CRP[2-4].
Authors' answer: Based on current research it has been reported that despite low or normal values of CRP in AS and PsA patient’s disease is considered as active. Recent studies, such as Sokolova et al. [1], presented that similarly to our study, inflammatory mediators expression pattern is mostly related with disease phenotype. Also, in study by Brown et al. 110 out of 197 patients with moderate to severe AS enrolled into MEASURE 1 and MEASURE 2 studies had lower CRP than 10mg/l [2].
-> I suggest putting this paragraph in “Results” section.
According to reviewer’s suggestion paragraph on CRP, ESR levels has been added to the “Results” part
- Minor concerns;
- Please add footnotes, which explain the abbreviation and statistical analysis, in the bottom of each table and figure.
Authors' answer: Footnotes explaining abbreviation were added under figures and tables.
- The footnotes of all pictures and tables are not properly modified. Footnote should have statistical description (such as Pearson's or Spearman's correlations, etc.) and abbreviation to facilitate interpretation of contents in pictures or tables. In particular, footnote of Table 1 does not describe the meaning of p values.
Statistical description including type of statistical test, rho value, p value, chi2 p- value and abbreviations were added.
- Please state consistently below the decimal point of the data in the tables
Authors' answer: The manuscript has been reformatted to match reviewers’ concerns
- Unmodified descriptions are observed in Table 1. Please revise them further.
All data was revised and presented with a decimal fraction with a denominator of 10
